# OpenReview forum: "Risk-Aware Distributional Intervention Policies for Language Models"
_ICLR.cc/2025/Conference — Submitted to ICLR 2025_

### Official Review · Reviewer_vXi8 · 2024-10-23

**Soundness:** 3
**Presentation:** 3
**Contribution:** 3
**Rating:** 6
**Confidence:** 4

**Summary:**

This paper proposes a method for detecting and mitigating undesirable content generation, which includes a set of classifiers and an intervention strategy. The intervention strategy is derived by the authors based on statistical theory and has shown promising results on the LLaMA model.

**Strengths:**

1. This paper proposes a novel intervention strategy to mitigate undesirable content generation, which is a significant research problem.
2. The authors propose three criteria for layer-wise intervention and conduct corresponding mathematical modeling and statistical analysis.
3. The authors demonstrate the effectiveness of this method through experiments.

**Weaknesses:**

The derivation of the intervention method in this paper is based on the assumption that distributions P and Q follow a normal distribution. Can the authors verify that the activation values sampled at each layer in LLMs indeed follow a normal distribution? If P and Q follow other distributions, can the method in this paper be correspondingly adapted?

**Questions:**

1. When combining Few-shot prompting with the RADIANT method, will the accuracy of the classifier obtained in the first stage change? Is it necessary to retrain the classifier?
2. How does the accuracy of the first-stage classifier affect the effectiveness of the second-stage intervention method? The paper uses a Linear Logistic Classifier as the classifier in the first stage; if a more accurate classifier is used, can the intervention effectiveness be further improved?
3. As shown in Figure 1 of the paper, there are significant differences in classifier accuracy across different model layers. Is it possible to only select certain model layers for the second-stage intervention?

---

> ### Author Response · Authors · 2024-11-23
> **Thank you for your valuable feedback and appreciation of our work**
>
> We thank the reviewer for the valuable feedback and appreciation of our work. Here are our replies to your questions.
>
> > The derivation of the intervention method in this paper is based on the assumption that distributions P and Q follow a normal distribution. Can the authors verify that the activation values sampled at each layer in LLMs indeed follow a normal distribution?
>
> We have used two types of normality testing (univariate and multivariate) for the attention heads that are classified as undesirable ($\hat{P}$) and after applying intervention ($Q$) -- the results are not clear: the univariate returns normality most of the time with good p-values, but a more realistic test, the Henze-Zirkler test for multivariate normality, usually dismisses the null-hypothesis (means there is no conclusion that the distribution is normal).
>
> > If P and Q follow other distributions, can the method in this paper be correspondingly adapted?
>
> If P and Q follow other distributions, we can still use the optimization to solve for the linear intervention. However, the probability guarantee will not be met all the time. We note that without putting the assumption on the high-dimensional distribution P and Q, the optimization problem will be ill-posed and, therefore, very hard to solve. For example, we might need to estimate P and Q via non-parametric methods. The linear intervention is also an optimal transport plan when using Gaussian assumptions.
>
> > When combining Few-shot prompting with the RADIANT method, will the accuracy of the classifier obtained in the first stage change? Is it necessary to retrain the classifier?
>
> Indeed, we need to retrain the classifier, and combining few-shot prompting with the RADIANT method will result in a change in the accuracy of the classifier obtained in the first stage.
>
> > How does the accuracy of the first-stage classifier affect the effectiveness of the second-stage intervention method? The paper uses a Linear Logistic Classifier as the classifier in the first stage; if a more accurate classifier is used, can the intervention effectiveness be further improved?
>
> The accuracy of the first-stage classifier significantly impacts the effectiveness of the second-stage intervention method. This is because a highly accurate classifier minimizes the false-negative rate, ensuring that undesirable text is effectively detected, and reduces the false-positive risk, preventing desirable text from being misclassified as undesirable. This is crucial to preserving the original semantics of desirable text while effectively intervening in undesirable cases.
>
> In our work, we use a Logistic Classifier in the first stage with the training loss proposed in Section 2 and the subsequent distributional intervention that can be efficiently optimized using semidefinite programming. Moreover, our method demonstrates superior performance compared to other approaches, such as the Non-Linear Inference Time Intervention (NL-ITI) (Hoscilowicz et al., 2024), which employs an MLP classifier. We added this result to Table 1 of the updated PDF.
>
> | Model           | Methods              |   True * Info (%) ↑ |   True (%) ↑ |   MC1 ↑ |   MC2 ↑ |   CE ↓ |   KL ↓ |
> |:----------------|:---------------------|--------------------:|-------------:|--------:|--------:|-------:|-------:|
> | Llama-7B        | NL-ITI               |               29.06 |        38.04 |   34.02 |   45.69 |   2.19 |   0.07 |
> |                 | RADIANT (ours)       |               40.36 |        44.48 |   41.22 |   48.97 |   2.20 |   0.08 |
> |                 | FSP + NL-ITI         |               45.97 |        47.31 |   38.37 |   46.95 |   2.20 |   0.07 |
> |                 | FSP + RADIANT (ours) |               49.31 |        57.43 |   41.97 |   55.31 |   2.20 |   0.08 |
> |                 |
> | Llama-3-8B        | NL-ITI               |               35.98 |        45.72 |   45.02 |   50.32 |   2.50 |   0.13 |
> |                 | RADIANT (ours)       |               37.78 |        50.82 |   43.82 |   52.98 |   2.48 |   0.15 |
> |                 | FSP + NL-ITI         |               40.70 |        46.03 |   34.06 |   48.41 |   2.49 |   0.14 |
> |                 | FSP + RADIANT (ours) |               44.09 |        52.02 |   37.98 |   54.61 |   2.52 |   0.15 |
> |                 |
> | Llama2-chat-13B | NL-ITI               |               57.13 |        60.82 |   39.30 |   57.24 |   2.33 |   0.17 |
> |                 | RADIANT (ours)       |               63.68 |        74.20 |   39.95 |   58.18 |   2.35 |   0.18 |
> |                 | FSP + NL-ITI         |               59.62 |        61.77 |   42.15 |   55.87 |   2.36 |   0.17 |
> |                 | FSP + RADIANT (ours) |               64.68 |        67.75 |   42.52 |   59.99 |   2.38 |   0.18 |

---

> > ### Author Response · Authors · 2024-11-23
> > **Response to Reviewer vXi8 (2/2)**
> >
> > > As shown in Figure 1 of the paper, there are significant differences in classifier accuracy across different model layers. Is it possible to only select certain model layers for the second-stage intervention?
> >
> > It is certainly possible to select certain model layers for the 2nd-stage intervention – our framework allows for flexibility. However, we might face different issues from the current framework, most notably a more computational bottleneck. Combining interventions across multiple layers will be a direction for our future work.

---

> > > ### Comment · Reviewer_vXi8 · 2024-11-26
> > >
> > > Thank you to the authors for their response. I have no further questions.

---

> > > > ### Author Response · Authors · 2024-11-26
> > > >
> > > > Dear Reviewer vXi8,
> > > >
> > > > Thank you for your response.
> > > >
> > > > Best regards,
> > > >
> > > > The authors.

---

### Official Review · Reviewer_CbQ1 · 2024-10-27

**Soundness:** 3
**Presentation:** 2
**Contribution:** 2
**Rating:** 5
**Confidence:** 5

**Summary:**

The paper introduces RADIANT, a novel two-stage approach for detecting and mitigating undesirable content in language models. The method involves training layerwise classifiers to identify harmful outputs and employing headwise distributional interventions to adjust model activations. RADIANT offers a resource-efficient solution with probabilistic guarantees for effectiveness.

**Strengths:**

The paper introduces a novel two-stage method, RADIANT, for detecting and mitigating undesirable content in language models. This method combines layerwise classifiers and headwise interventions.

The method is empirically validated on several benchmarks, including TruthfulQA, demonstrating its effectiveness compared to existing baselines.

**Weaknesses:**

1. The proposed method, RADIANT, appears to offer minimal modifications compared to the Inference-Time Intervention (ITI) method. The main difference lies in the selection of attention heads between RADIANT and ITI. This slight variation may not constitute a significant advancement over existing methods.

2. The paper does not sufficiently compare RADIANT with other related methods in the field. The following works should be considered for comparison:

[1] Truth Forest: Toward Multi-Scale Truthfulness in Large Language Models through Intervention without Tuning

[2] Non-Linear Inference Time Intervention: Improving LLM Truthfulness

[3] Enhanced Language Model Truthfulness with Learnable Intervention and Uncertainty Expression

3. The paper lacks results from Supervised Fine-Tuning (SFT) as a baseline, which is important for benchmarking. Referring to the experimental settings used in ITI and including SFT results would allow for a more comprehensive comparison.

3. A major limitation is the narrow focus on a limited number of datasets. Most experiments are conducted on the TruthfulQA dataset, with minimal generalization to only two additional datasets. This raises concerns about whether the effectiveness of RADIANT is universally applicable or specific to TruthfulQA. More tests on a variety of datasets would strengthen the paper.

4. The method for determining the crucial hyper-parameter remains ambiguous. It's unclear whether the hyper-parameters $\alpha$ and $ \Gamma $ are universally applicable or requires case-by-case optimization. Considering the improvement provided by RADIANT, selecting an appropriate hyper-parameter is vital, as suggested by Table 6 and Table 7.  However, the paper fails to explain how this selection should be done.

5. It seems that the method may be limited to the LLaMA family of models. Conducting experiments on LLMs from other families could demonstrate the broader applicability of RADIANT.

6. RADIANT shows an increase in CE and KL divergence compared to ITI, suggesting that the method deviates from the initial distribution. This could indicate potential adverse effects on the model's performance on other tasks. The paper should address this concern and discuss how RADIANT impacts overall model performance.

**Questions:**

1. What are the cost differences between ITI and RADIANT as presented in Table 5?

2. How is the difference between generation accuracy and probe accuracy used in the evaluation?

3. The equations and symbols presented in the paper can be challenging to follow without clear explanations of each component and its role in the method. Providing detailed descriptions and context would improve readability and comprehension.

---

> ### Author Response · Authors · 2024-11-22
> **Thank you for your valuable feedback and appreciation of our work**
>
> > The proposed method, RADIANT, appears to offer minimal modifications compared to the Inference-Time Intervention (ITI) method. The main difference lies in the selection of attention heads between RADIANT and ITI. This slight variation may not constitute a significant advancement over existing methods.
>
> We thank the reviewer for raising the concern, but we argue that this is not the case. In fact, besides the difference in the selection of attention heads between RADIANT and ITI (as the reviewer has pointed out), we proposed a whole new way of finding the intervention value $\Delta$ for each of the selected attention heads. This is done by solving a novel optimization problem -- a stochastic program inspired by optimal transport theory. While involving technical details, the overall objective of this optimization problem is to be effective in converting the undesirable activations to the desirable regions, while minimizing the magnitude of the intervention to sustain the context of the input. For this fact, we firmly believe that this is a stark contrast with ITI in terms of finding each of the head's intervention value.
>
> > A major limitation is the narrow focus on a limited number of datasets. The paper does not sufficiently compare RADIANT with other related methods in the field. It seems that the method may be limited to the LLaMA family of models.
> >
> > The paper lacks results from Supervised Fine-Tuning (SFT) as a baseline, which is important for benchmarking. Referring to the experimental settings used in ITI and including SFT results would allow for a more comprehensive comparison.
>
> Thank you for the two valuable suggestions. We have conducted a comprehensive set of experiments, incorporating additional baselines and a new dataset focused on toxicity generation. For detailed information, we kindly direct the reviewer to our shared response. Below, we provide a brief summary of our efforts to address your concerns:
>
> For the QA task, we included two recent baselines, NL-ITI (Hoscilowicz et al., 2024) and LITO (Bayat et al., 2024), which employ approaches similar to ITI. These comparisons can be found in our shared response or in Table 1 of the revised paper. Overall, RADIANT demonstrates superior performance compared to NL-ITI and LITO across three base models—Llama-7B, Llama3-8B, and Llama2-chat-13B—on nearly all metrics, except a few cases involving MC metrics.
>
> As you suggested, we also expanded our experiments to evaluate RADIANT’s effectiveness on GPT and Gemma base models. The results show that RADIANT consistently outperforms finetuning-free methods in terms of truthfulness and informativeness. Further details on these results are available in our shared response or Appendix A.3 of the revised paper
>
> We investigated RADIANT’s performance on a different task: toxicity mitigation. In this experiment, we compared RADIANT with various state-of-the-art methods, including MiMiC [2] (both Mean Matching and Mean+Covariance Matching versions), DEXPERTS [3], DAPT [4], UDDIA [5], PPLM [6], and GOODTRIEVER [7]. These baselines were categorized into two groups: **methods requiring extensive fine-tuning or gradient-based computations during inference (e.g., DAPT, GeDI, PPLM, UDDIA, DEXPERTS, GOODTRIEVER)** and lightweight inference-time methods (e.g., MiMiC, ITI, RADIANT). While the fine-tuning group generally performs better on toxicity metrics, these methods come with high computational costs due to the need for fine-tuning or gradient computations. In contrast, inference-time methods like MiMiC, ITI, and RADIANT offer comparable toxicity reduction but are significantly more resource-efficient.
> Among these, RADIANT achieves the best toxicity reduction while maintaining high fluency and diversity. Notably, RADIANT outperforms almost all methods in fluency (except UDDIA) and matches the diversity of top-performing methods like PPLM. This underscores RADIANT’s ability to balance efficiency and performance effectively. For a detailed discussion of the experimental setup, metrics, evaluation framework, and baseline results, we encourage the reviewer to consult our shared response or Appendix A.4 of the revised paper.

---

> ### Author Response · Authors · 2024-11-22
> **Response to Reviewer CbQ1 (2/3)**
>
> > The method for determining the crucial hyper-parameter remains ambiguous. It's unclear whether the hyper-parameters $\alpha$ and $\Gamma$ are universally applicable or require case-by-case optimization. Considering the improvement provided by RADIANT, selecting an appropriate hyper-parameter is vital, as suggested by Table 6 and Table 7. However, the paper fails to explain how this selection should be done.
>
> We apologize for the ambiguity in selecting $\alpha$ and $\Gamma$ in the main paper. We argue that similar to many of the machine learning frameworks and algorithms, having a fixed set of $\alpha$ and $\Gamma$ that can be universally applicable is highly improbable (think bias-variance tradeoff in model selection). Therefore, we suggest using the common practice in both academic and industrial settings, which is grid search combined with cross-validation.
>
> > RADIANT shows an increase in CE and KL divergence compared to ITI, suggesting that the method deviates from the initial distribution. This could indicate potential adverse effects on the model's performance on other tasks. The paper should address this concern and discuss how RADIANT impacts overall model performance.
>
> While it is true that we observed a general increase in cross-entropy (CE) and Kullback-Leibler (KL) divergence when comparing RADIANT to ITI, the increases are typically not substantial, with differences ranging from 0.01 to 0.03. Furthermore, we found that higher True*Info and True metrics often correlate with more desirable qualitative outputs (as detailed in Appendix B), where our framework consistently outperforms ITI.
>
> > What are the cost differences between ITI and RADIANT as presented in Table 5?
>
> | Base Models         | ITI  | RADIANT | RADIANT-P |
> |---------------------|------|---------|-----------|
> | Gemma-2B           | 2.53 | 6.82    | 1.75      |
> | GPT-2 Large        | 2.43 | 3.01    | 1.65      |
> | Llama-7B           | 2.46 | 3.09    | 1.45      |
> | Llama3-8B          | 2.51 | 3.32    | 1.55      |
> | Llama2-chat-13B    | 2.51 | 4.72    | 1.57      |
>
> From the theoretical aspect, it is obvious that a head intervention of ITI, which is just a vector addition, is faster than that of RADIANT, which comprises a matrix multiplication and a matrix addition operator. The table above reports the average percentage increase in inference time per answer of ITI and RADIANT across the base models. It is observed that the normal version of RADIANT imposes more additional time in inference than ITI does. However, it is worth noting that all interventions of RADIANT are conducted on the same layer, while those of ITI are on multiple pairs of layer heads. This attribute of RADIANT allows us to parallel the interventions, which is impossible for ITI. We denoted the parallel version of RADIANT as RADIANT-P. RADIANT-P offers the same decent results as RADIANT but imposes much less computation cost into base models than RADIANT and ITI. We also included this table in the updated PDF.
>
> > How is the difference between generation accuracy and probe accuracy used in the evaluation?
>
> Generation accuracy refers to the performance on QA tasks (such as multiple-choice questions in TruthfulQA), while probe accuracy refers to the performance of the probing method described in Section 2 of our paper.
>
> > The equations and symbols presented in the paper can be challenging to follow without clear explanations of each component and its role in the method. Providing detailed descriptions and context would improve readability and comprehension.
>
> Thank you for this constructive comment on our presentation. In our revised version, we added a paragraph outlining our intuition at the beginning of Section 3. We also revised the three beginning paragraphs of Section 3 to make it less notation-dense.
>
>
> **Reference**
>
> [1] https://huggingface.co/openai-community/gpt2-large
>
> [2] Singh, Shashwat, et al. "Representation Surgery: Theory and Practice of Affine Steering." Forty-first International Conference on Machine Learning.
>
> [3] Liu, Alisa, et al. "DExperts: Decoding-time controlled text generation with experts and anti-experts." arXiv preprint arXiv:2105.03023 (2021).
>
> [4] Gururangan, Suchin, et al. "Don't stop pretraining: Adapt language models to domains and tasks." arXiv preprint arXiv:2004.10964 (2020).
>
> [5] Yang, Zonghan, et al. "Unified detoxifying and debiasing in language generation via inference-time adaptive optimization." arXiv preprint arXiv:2210.04492 (2022).
>
> [6] Dathathri, Sumanth, et al. "Plug and play language models: A simple approach to controlled text generation." arXiv preprint arXiv:1912.02164 (2019).
>
> [7] Pozzobon, Luiza, et al. "Goodtriever: Adaptive toxicity mitigation with retrieval-augmented models." arXiv preprint arXiv:2310.07589 (2023).

---

> > ### Author Response · Authors · 2024-11-23
> > **Response to Reviewer CbQ1 (3/3)**
> >
> > > What are the cost differences between ITI and RADIANT as presented in Table 5?
> >
> > | Base Models         | ITI  | RADIANT | RADIANT-P |
> > |---------------------|------|---------|-----------|
> > | Gemma-2B           | 2.53 | 6.82    | 1.75      |
> > | GPT-2 Large        | 2.43 | 3.01    | 1.65      |
> > | Llama-7B           | 2.46 | 3.09    | 1.45      |
> > | Llama3-8B          | 2.51 | 3.32    | 1.55      |
> > | Llama2-chat-13B    | 2.51 | 4.72    | 1.57      |
> >
> > From the theoretical aspect, it is obvious that a head intervention of ITI, which is just a vector addition, is faster than that of RADIANT, which comprises a matrix multiplication and a matrix addition operator. The table above reports the average percentage increase in inference time per answer of ITI and RADIANT across the base models. It is observed that the normal version of RADIANT imposes more additional time in inference than ITI does. However, it is worth noting that all interventions of RADIANT are conducted on the same layer, while those of ITI are on multiple pairs of layer heads. This attribute of RADIANT allows us to parallel the interventions, which is impossible for ITI. We denoted the parallel version of RADIANT as RADIANT-P. RADIANT-P offers the same decent results as RADIANT but imposes much less computation cost into base models than RADIANT and ITI. We also included this table in the updated PDF.
> >
> > > How is the difference between generation accuracy and probe accuracy used in the evaluation?
> >
> > Generation accuracy refers to the performance on QA tasks (such as multiple-choice questions in TruthfulQA), while probe accuracy refers to the performance of the probing method described in Section 2 of our paper.

---

> > > ### Comment · Reviewer_CbQ1 · 2024-11-25
> > >
> > > Thank you for your clarifications.
> > >
> > > Compared to ITI, this method is an incremental approach with limited innovation.
> > >
> > > Incorporating results from SFT methods,  would provide a performance ceiling for models. It should be included for comparison.
> > >
> > > Another concern is that this method may require extensive hyperparameter tuning.
> > >
> > > I am maintaining my current score.

---

> > > > ### Author Response · Authors · 2024-11-26
> > > > **Response to Reviewer CbQ1's additional comments**
> > > >
> > > > We thank the reviewer for engaging in the discussion. Following are our responses, and please feel free to ask additional questions.
> > > >
> > > > > Compared to ITI, this method is an incremental approach with limited innovation.
> > > >
> > > > We have provided in [the rebuttal to your review](https://openreview.net/forum?id=tLPHgQMw08&noteId=hyz1TyoFzx) a long argument on our contribution of the paper. We restated here that there are two fundamental differences compared with ITI:
> > > >
> > > > 1. The risk-aware methodology for training the linear probe of attention heads; the decision to intervene on attention heads that belong to a single layer instead of attention heads across layers.
> > > > 2. A whole new and novel optimization problem to find the optimal intervention values based on the three criteria defined in the paper. This is a fundamental difference compared to the heuristic rule in the ITI paper.
> > > >
> > > > With all due respect, we think the reviewer was stating a subjective opinion on the contribution of our work. **We kindly ask the reviewer to make a more objective counter-argument on which of the two parts above the reviewer thinks are incremental.**
> > > >
> > > > > Incorporating results from SFT methods, would provide a performance ceiling for models. It should be included for comparison.
> > > >
> > > > We are currently running this comparison, thanks to the extension of the discussion period. We will report on this in the upcoming days.
> > > >
> > > > > Another concern is that this method may require extensive hyperparameter tuning.
> > > >
> > > > Hyperparameter tuning is a fundamental part of both research and applied Machine Learning nowadays. Comparing to Supervised Fine-Tuning that also needs hyperparameters tuning (most importantly learning rate, but also some plethora of others such as optimizer type, learning rate scheduler, LoRA parameters if applicable, sequence length, batch size), the usual computational cost of hyperparameter tuning for finetuning-free methods such as ours or ITI is several time smaller, since we only run models in inference mode.
> > > >
> > > > Finally, we note that the state-of-the-art ITI also requires an extensive amount of hyperparameter optimization to work properly, of which the reviewer can check at the [ITI's result replication table.](https://github.com/likenneth/honest_llama/blob/master/validation/iti_replication_results.md)

---

> > > > > ### Author Response · Authors · 2024-11-29
> > > > > **Re: Comparison with Supervised Fine-Tuning (SFT)**
> > > > >
> > > > > Dear Reviewer CbQ1,
> > > > >
> > > > > Following your suggestion, we provide below the results for supervised fine-tuning comparison benchmark. We hope that this has resolved your concern about the lack of SFT comparison.
> > > > >
> > > > > **SFT on GPT2-Large**
> > > > >
> > > > > Due to computational constraints and SFT's requirement to finetune all LLM parameters that demands substantial GPU resources for backpropagation operation, we can only perform SFT on the GPT2-large, the smallest model in our experiments. The results are available in Table 1 below. Once again, we want to highlight the advantages of inference time methods like ours: by avoiding gradient computation or backpropagation, they offer a lightweight, fast, versatile, and economical way to improve the performance of LLMs. This is especially useful in low-resource scenarios.
> > > > >
> > > > > *Table 1: Quantitative results of different intervention methods on TruthfulQA dataset on GPT2-Large. Parameters of RADIANT: $\alpha = 2.5, \Gamma = 15$.*
> > > > > | Methods | True * Info (%) ↑ | True (%) ↑ | MC1 ↑ | MC2 ↑ | CE ↓ | KL ↓ |
> > > > > |---------|------------------|-------------|--------|--------|-------|-------|
> > > > > | Unintervened | 19.20 | 21.91 | 23.57 | 40.75 | 2.8 | 0.0 |
> > > > > | SFT | **35.16** | 38.28 | **35.70** | **53.57** | 3.27 | 0.46 |
> > > > > | ITI | 26.94 | 31.09 | 24.68 | 42.31 | 2.94 | 0.13 |
> > > > > | FSP | 21.82 | 27.30 | 25.34 | 42.07 | 2.8 | 0.0 |
> > > > > | RADIANT (ours) | 30.18 | **38.73** | 25.14 | 42.14 | 2.92 | 0.12 |
> > > > > | FSP + ITI | 29.53 | 30.45 | 25.12 | **44.79** | 2.98 | 0.18 |
> > > > > | FSP + RADIANT (ours) | **35.36** | **40.41** | **26.18** | 44.29 | 2.94 | 0.16 |
> > > > >
> > > > > **SFT on Llama-7B**
> > > > >
> > > > > Because Llama-7B is used as a base model for many of our experiments, we also include the results SFT on Llama-7B for comparison, but it is worth noting that this result is referred from the ITI paper (Table 1, Li et al. 2024). Since our evaluation framework differs from ITI in terms of the GPT-judge and GPT-info models (which is attributed to the fact that these models in the ITI paper are no longer available in the OpenAI), the results may not be fair for comparison.
> > > > >
> > > > > *Table 2: Quantitative results of different intervention methods on TruthfulQA dataset on Llama-7B. Parameters of RADIANT: $\alpha = 2.5, \Gamma = 15$.*
> > > > > | Methods | True * Info (%) ↑ | True (%) ↑ | MC1 ↑ | MC2 ↑ | CE ↓ | KL ↓ |
> > > > > |---------|------------------|-------------|--------|--------|-------|-------|
> > > > > | Unintervened | 21.15 | 22.16 | 25.58 | 40.54 | 2.13 | 0.00 |
> > > > > | SFT* | 36.10 | **47.10** | 24.20 | - | 2.10 | 0.01 |
> > > > > | ITI | 26.52 | 28.03 | 27.78 | 43.59 | 2.20 | 0.07 |
> > > > > | FSP | 36.13 | 39.78 | **34.03** | **50.34** | 2.13 | 0.00 |
> > > > > | RADIANT (ours) | **40.36** | **44.48** | 30.91 | 46.13 | 2.19 | 0.07 |
> > > > > | FSP + ITI | 40.63 | 45.16 | 35.50 | 52.48 | 2.20 | 0.07 |
> > > > > | FSP + RADIANT (ours) | **49.31** | **57.43** | 37.97 | **55.31** | 2.20 | 0.08 |
> > > > >
> > > > > **From Table 2, SFT achieves the best performance in terms of MC metrics and reaches a high score of True * Info and True. Regarding the True score, RADIANT still outperforms SFT in both the individual and integrating versions with FSP, offering 38.73% and 40.41% correct answers, respectively. When combined with FSP, RADIANT achieves 35.36% in True * Info score, surpassing SFT but requiring much less resources.**
> > > > >
> > > > > For the implementation of SFT, we use the SFTTrainer framework from Hugging Face, one of the most popular frameworks for this algorithm. While we kept the default for many parameters proposed by the library, we had to tune almost all of the important parameters like learning rate, parameters of Adam optimizer, weight decay, and so on, to get a consistent and stable fine-tuned model. Some important parameters for SFT are reported in the table below, while its best performance is represented in Table 2.
> > > > >
> > > > > This observation strongly supports the practicability of RADIANT, which only necessitates tuning two key hyper-parameters $\alpha$ in the probe loss, and $\Gamma = \Phi^{-1}(1-\gamma)$ in the computation of the intervention map. A thorough analysis of these parameters in an attempt to offer insights into their impact is presented in Appendix A.
> > > > >
> > > > > Furthermore, compared to other methods in the field, like ITI, we claim that the grid search on two hyper-parameters like ours is efficient and reasonable, so it is not harder to tune the hyper-parameters of RADIANT than other previous works.
> > > > >
> > > > > *Table 3: parameter for supervised fine-tuning*
> > > > > | Parameter | Value |
> > > > > |-----------|-------|
> > > > > | learning_rate | 0.00002 |
> > > > > | weight_decay | 0 |
> > > > > | adam_beta1 | 0.8 |
> > > > > | adam_beta2 | 0.999 |
> > > > > | adam_epsilon | $$1 \times 10^{-8}$$ |
> > > > > | max_grad_norm | 1 |
> > > > > | batch_size | 32 |
> > > > > | epochs_num | 5 |
> > > > > | lr_scheduler_type | linear |

---

> > > > > > ### Author Response · Authors · 2024-12-02
> > > > > >
> > > > > > Dear Reviewer CbQ1,
> > > > > >
> > > > > > Thank you again for your constructive feedback. Once again, we respond to your main concern about the motivations and distinctions of our paper vs. ITI (Li et al. 2024). We note that this is a summary of what we have written already in our paper. We hope that the reviewer can reconsider raising their current evaluation of our paper.
> > > > > >
> > > > > > Previous works like ITI and LOFiT intervene on multiple heads across multiple layers. This approach creates a mismatch between pre-intervention activations during training and inference. For instance, in ITI, the activations to construct interventions of any head in layer l are collected when no intervention occurs in layers 0 to l-1. However, in the inference phase, the interventions happen in layers 0 to l-1, which causes a distribution shift of activations between training and inference, hindering the performance of interventions. **To accommodate this issue, we proposed intervening on all heads of a layer instead and proposed layer-wise probes using the voting rule to increase accuracy further. This refers to Section 2 of our paper.**
> > > > > >
> > > > > > Another motivation for intervening on one layer is allowing us to do intervention parallelly. All interventions of RADIANT are conducted on the same layer, while those of ITI are on multiple pairs of layer heads. This attribute of RADIANT allows us to parallel the interventions, which is impossible for ITI. We denoted the parallel version of RADIANT as RADIANT-P. RADIANT-P offers the same decent results as RADIANT but imposes much less computation cost into base models than RADIANT and ITI. The table below reports the average percentage increase in inference time per answer of ITI and RADIANT across the base models. We also included this table in the revised PDF.
> > > > > >
> > > > > > | Base Models         | ITI  | RADIANT | RADIANT-P |
> > > > > > |---------------------|------|---------|-----------|
> > > > > > | Gemma-2B           | 2.53 | 6.82    | 1.75      |
> > > > > > | GPT-2 Large        | 2.43 | 3.01    | 1.65      |
> > > > > > | Llama-7B           | 2.46 | 3.09    | 1.45      |
> > > > > > | Llama3-8B          | 2.51 | 3.32    | 1.55      |
> > > > > > | Llama2-chat-13B    | 2.51 | 4.72    | 1.57      |
> > > > > >
> > > > > > The proposal of the risk-aware loss based on the parameters $\alpha$ is essential in real-world scenarios where we want LLMs to be more conservative in sensitive applications like in medical or legal sectors. In these cases, LLMs should avoid providing uncertain or wrong information, so high values of $\alpha$ are recommended. This decent feature of RADIANT is elaborated on in Appendix A.
> > > > > >
> > > > > > **Our main and novel contribution is the construction of the head-wise intervention through solving an intuitive optimization problem.** Previous works like ITI do not guarantee two things.
> > > > > >
> > > > > > First, they don’t concern the semantic shifts of the answers. They use a heuristic intervention, which is a vector addition operator that potentially shifts the activations too far from the original ones. This likely leads to semantic changes in answers. RADIANT addresses this problem by directly minimizing the distributional shift of activations compared to its original state, which is clearly observed in Problem (3).
> > > > > >
> > > > > > Second, they are not concerned about whether the post-intervention activations lie in desirable areas. We solve this issue by establishing a probabilistic guarantee for the extent of post-intervention activations in the desirable areas. This is incorporated in the constraint of Problem (3).
> > > > > >
> > > > > > With such improvements from both practical and theoretical aspects, our proposed methods outperform other baselines, which is apparent from empirical experiments.
> > > > > >
> > > > > > **Counting all significant contributions listed above, we respectfully disagree with your opinion that our paper just follows the ITI paper. Our paper not only indicates serious problems in existing intervention works but also proposes effective ways to accommodate them.**
> > > > > >
> > > > > > Best regards, The authors.

---

### Official Review · Reviewer_KLQW · 2024-10-28

**Soundness:** 3
**Presentation:** 1
**Contribution:** 2
**Rating:** 3
**Confidence:** 4

**Summary:**

Language models are prone to occasionally undesirable generations. This paper proposes a two-stage framework RADIANT to improve the language model output in the inference stage. In the first stage, RADIANT trains an ensemble of layer-wise classifiers to detect undesirable content using activations by probing the language model. In the second stage, for contents that are detected as undesirable, the method proposes layer-wise distributional intervention policies that perturb the attention heads minimally with a simple linear map.

In this paper, various experiments and comparisons on TruthfulQA are conducted to verify the effectiveness of the proposed method RADIANT.

**Strengths:**

The strengths of this paper can be listed as follows:
1. The optimal headwise interventions method proposed in this paper is motivating. The method has achieved a good performance both in theory and experiments.
2. The paper has conducted detailed experiments on TruthfulQA dataset to compare RADIANT with different kinds of baselines.

**Weaknesses:**

The weaknesses of this paper can be listed as follows:
1. **Bad paper writing.** I think there exists a big problem in paper writing. The organization of the paper is rather confusing. For example, some settings in Section 2 can be explained in the experimental settings in Section 4 or Appendix. I also don’t like the description in Section 3. The paper puts together too many symbolic definitions, which takes me a lot of time to understand. In addition, the presentations of the experimental results in table 1, table 2 and table 3 are redundant. The information can be summarized in one table instead.
2. **Limited experiment datasets.** The experiments in the paper mostly focus on TruthfulQA without considering other tasks to show the effectiveness of RADIANT on other datasets. As the paper discussed controllable text generation, I think many tasks in controllable text generation can be used to conduct experiments, such as sentiment controllable generation, detoxification, etc.
3. **Limited baselines.** The baselines listed in this paper are limited. I think this type of method like RADIANT that intervene the generation stage needs to be compared with the controllable decoding method, for example, Dexperts (https://aclanthology.org/2021.acl-long.522/).

**Questions:**

See the weaknesses above.

---

> ### Author Response · Authors · 2024-11-22
> **Thank you for your valuable feedback and appreciation of our work**
>
> We thank the reviewer for the valuable feedback and appreciation of our work. Here are our replies to your questions.
>
> >  I think there exists a big problem in paper writing. For example, some settings in Section 2 can be explained in the experimental settings in Section 4 or Appendix. I also don’t like the description in Section 3. The paper puts together too many symbolic definitions, which takes me a lot of time to understand.
>
> Thank you for this constructive comment on our presentation. In our revised version, we added a paragraph outlining our intuition at the beginning of Section 3. We also revised the three beginning paragraphs of Section 3 to make it less notation-dense.
>
> However, we argue that the placement of problem settings in Section 2 is intentional and crucial for clarity. Section 2 outlines the problem statement, providing the essential context for understanding the proposed method. This includes a description of the language model's architecture, the notation for activations at each layer, and the goal of detecting and modifying undesirable text. Moving this information would disrupt the logical flow of the paper and make it harder for readers to grasp the foundational concepts before diving into the specific methods and experimental details.
>
> The reviewer criticized Section 3 for its use of symbolic definitions, finding them difficult to understand. While acknowledging the complexity, we argue that these definitions are necessary to present the novel method, RADIANT, with the required precision.
>
> We aim to provide a rigorous justification and explanation for our method, so we make Section 3 technical. We also provide a clear definition of each technical term to avoid any confusion for the reader. Moreover, the notations we use are relatively standard and widely used in optimization and optimal transport, and many papers published at ICLR use similar notations.
>
> We think that posing the approximation of the intervention as an optimization problem (more particularly a semi-definite program) is rigorous and necessary for a scientific ML venue such as ICLR. We think the current iteration of the section is as optimally simple as possible, as moving all the derivations to the appendix will make the paper less scientifically rigorous.
>
> Moreover, to help alleviating the potential difficulty of some readers, in Section 3, we have included visual aids, such as Figure 2, which illustrates the effect of headwise intervention,
>
> >  The presentations of the experimental results in table 1, table 2 and table 3 are redundant. The information can be summarized in one table instead.
>
> The three tables describe three different tasks, separated by Subsections 4.2.1, 4.2.2, and 4.3. Each table showcases the effectiveness of RADIANT in a different aspect. Table 1 benchmarks the Llama 7-B model, which has always been used by previous works in the field, increasing our reliability and making a fair comparison to previous works. Table 2 benchmarks on the Llma3 8-B showcase the superiority of RADIANT on the newest model, unfolding broad applications of RADIANT on up-to-date models. Table 3 reported the results on a bigger base model, which shows that the effectiveness of RADIANT remains when scaling up the model size.

---

> > ### Author Response · Authors · 2024-11-22
> > **Response to Reviewer KLQW (2/2)**
> >
> > > Limited experiment datasets and limited baselines.  The experiments in the paper mostly focus on TruthfulQA without considering other tasks to show the effectiveness of RADIANT on other datasets. The baselines listed in this paper are limited.
> >
> > We have performed an extensive set of experimental benchmarks, which include more baselines and a different dataset on toxicity generation. We invite the reviewer to refer to our common response for more details. Briefly, we can summarize our work aiming to address your concerns as follows:
> >
> > In the QA task, we included two additional baselines NL-ITI (Hoscilowicz et al., 2024) and LITO (Bayat et al., 2024). These baselines are up-to-date works with similar approaches to ITI. We invite the reviewer to refer to our common response or Table 1 in our revised version to compare RADIANT and these methods. In general, our method outperforms NL-ITI and LITO in three base models, Llama-7B, Llama3-8B, and Llama2-chat-13B, regarding almost all metrics. The exception only happens in limited cases of MC metrics.
> >
> > We also extend the experiments to showcase the effectiveness of RADIANT on GPT and Gemma base models. It is observed that RADIANT is also superior to finetuning-free methods in terms of truthfulness and informativeness. The result of this experiment can be found in the common answer or Appendix A.3 of our revised version.
> >
> > Following your suggestions, we study the impact of RADIANT on a different task, which is toxic mitigation. In this experiment, we compare RADIANT with various up-to-date methods, including MiMiC [2], DEXPERTS [3], DAPT [4], UDDIA [5], PPLM [6], GOODTRIEVER [7]. As for MIMIC, we consider two versions: Mean Matching (MM) and Mean+Covariance Matching (MCM). Both these versions are introduced in their original paper. Baselines are divided into two groups: those requiring extensive fine-tuning or gradient computations during inference(e.g., DAPT, GeDI, PPLM, UDDIA, DExperts, GOODTRIEVER) and light-weight inference-time methods (e.g., MIMIC, ITI, RADIANT).
> >
> > While the fine-tuning group generally performs better on toxicity metrics, these methods are computationally expensive due to the necessity of fine-tuning or gradient computations during inference. Inference-time methods like MIMIC, ITI, and RADIANT achieve comparable toxicity reduction to fine-tuning methods but are significantly more resource-efficient. **Among these, RADIANT stands out by delivering the best toxicity reduction while also maintaining strong fluency and diversity. In fact, RADIANT’s fluency surpasses nearly all methods in the fine-tuning group except UDDIA, and its diversity is on par with the best (PPLM).** This highlights RADIANT’s balance of efficiency and performance. We highly suggest that the reviewer refer to our common answer or Appendix A.4 of our revised paper for detailed experiment setups, metrics, evaluation framework, and baseline results.
> >
> >
> > **Reference**
> >
> > [1] https://huggingface.co/openai-community/gpt2-large
> >
> > [2] Singh, Shashwat, et al. "Representation Surgery: Theory and Practice of Affine Steering." Forty-first International Conference on Machine Learning.
> >
> > [3] Liu, Alisa, et al. "DExperts: Decoding-time controlled text generation with experts and anti-experts." arXiv preprint arXiv:2105.03023 (2021).
> >
> > [4] Gururangan, Suchin, et al. "Don't stop pretraining: Adapt language models to domains and tasks." arXiv preprint arXiv:2004.10964 (2020).
> >
> > [5] Yang, Zonghan, et al. "Unified detoxifying and debiasing in language generation via inference-time adaptive optimization." arXiv preprint arXiv:2210.04492 (2022).
> >
> > [6] Dathathri, Sumanth, et al. "Plug and play language models: A simple approach to controlled text generation." arXiv preprint arXiv:1912.02164 (2019).
> >
> > [7] Pozzobon, Luiza, et al. "Goodtriever: Adaptive toxicity mitigation with retrieval-augmented models." arXiv preprint arXiv:2310.07589 (2023).

---

> ### Author Response · Authors · 2024-11-26
> **We are happy to provide further response should you have any request**
>
> Dear Reviewer KLQW,
>
> We have provided an extensive rebuttal that in our opinion has addressed your concerns. Could you at least please acknowledge that you have read it, and perhaps consider raising your evaluation of our work?
>
> Best regards,
>
> The authors.

---

> > ### Author Response · Authors · 2024-11-29
> > **Looking forward to hearing back from you**
> >
> > Dear Reviewer KLQW,
> >
> > **The extended rebuttal period of three weeks is coming to and end soon (on 2nd of December).** Once again, we think we have provided an extensive rebuttal that, in our opinion, has addressed your major concern regarding the empirical evaluations. Could you please take your time to read it and consider raising your evaluation?
> >
> > Best regards,
> >
> > The authors.

---

> > > ### Comment · Reviewer_KLQW · 2024-11-30
> > > **Official Response to Authors**
> > >
> > > I really appreciate the complementary experiments and the efforts made by the authors. The reponse by authors helps me correct some misunderstandings. However, I still have some concerns about the paper.
> > >
> > > 1. I think this paper still needs further revision. The authors mentioned in their responses that each table showcases the effectiveness of RADIANT in a different aspect. However, I believe that RADIANT is not targeted at a specific model, so it should be valid for at least 2-3 different LLMs to verify that this method works. Moreover, there can be more comparisons between experiments on different LLMs. For example, how the performance of RADIANT changes when the model size increases, and how other baselines behave. Therefore, I think the reasons provided by the authors do not convince me.
> > > 2. I do not see a clear motivation from this paper. Although some experiments in section 2 and section 3 show that there are some empirical rules in the latent space of language models, I don't think they are intuitive enough. I think there lacks a description of problems in existing methods. As stated by other reviewers, from the introduction of this paper, I would think this work just wants to follow ITI.
> > > 3. I think experiments provided in this paper lacks depth. I really appreciate the substantial experimental results provided by authors. However, there lacks further analysis and discussions.
> > >
> > > In conclusion, I think this work needs further revision.

---

> > > > ### Author Response · Authors · 2024-12-01
> > > > **Response to Reviewer KLQW (3/n)**
> > > >
> > > > > I do not see a clear motivation from this paper. Although some experiments in section 2 and section 3 show that there are some empirical rules in the latent space of language models, I don't think they are intuitive enough. I think there lacks a description of problems in existing methods. As stated by other reviewers, from the introduction of this paper, I would think this work just wants to follow ITI.
> > > >
> > > > We appreciate that the reviewer raised this concern. Let us elaborate on the problems in existing works and our respective solutions, which will clarify our contribution and intuition. We note that this is a summary of what we have written already in our paper.
> > > >
> > > > Previous works like ITI and LOFiT intervene on multiple heads across multiple layers. This approach creates a mismatch between pre-intervention activations during training and inference. For instance, in ITI, the activations to construct interventions of any head in layer l are collected when no intervention occurs in layers 0 to l-1. However, in the inference phase, the interventions happen in layers 0 to l-1, which causes a distribution shift of activations between training and inference, hindering the performance of interventions. **To accommodate this issue, we proposed intervening on all heads of a layer instead and proposed layer-wise probes using the voting rule to increase accuracy further. This refers to Section 2 of our paper.**
> > > >
> > > > Another motivation for intervening on one layer is allowing us to do intervention parallelly. All interventions of RADIANT are conducted on the same layer, while those of ITI are on multiple pairs of layer heads. This attribute of RADIANT allows us to parallel the interventions, which is impossible for ITI. We denoted the parallel version of RADIANT as RADIANT-P. RADIANT-P offers the same decent results as RADIANT but imposes much less computation cost into base models than RADIANT and ITI. The table below reports the average percentage increase in inference time per answer of ITI and RADIANT across the base models. We also included this table in the revised PDF.
> > > >
> > > > | Base Models         | ITI  | RADIANT | RADIANT-P |
> > > > |---------------------|------|---------|-----------|
> > > > | Gemma-2B           | 2.53 | 6.82    | 1.75      |
> > > > | GPT-2 Large        | 2.43 | 3.01    | 1.65      |
> > > > | Llama-7B           | 2.46 | 3.09    | 1.45      |
> > > > | Llama3-8B          | 2.51 | 3.32    | 1.55      |
> > > > | Llama2-chat-13B    | 2.51 | 4.72    | 1.57      |
> > > >
> > > > The proposal of the risk-aware loss based on the parameters $\alpha$ is essential in real-world scenarios where we want LLMs to be more conservative in sensitive applications like in medical or legal sectors. In these cases, LLMs should avoid providing uncertain or wrong information, so high values of $\alpha$ are recommended. This decent feature of RADIANT is elaborated on in Appendix A.
> > > >
> > > > **Our main and novel contribution is the construction of the head-wise intervention through solving an intuitive optimization problem.** Previous works like ITI do not guarantee two things.
> > > >
> > > > First, they don’t concern the semantic shifts of the answers. They use a heuristic intervention, which is a vector addition operator that potentially shifts the activations too far from the original ones. This likely leads to semantic changes in answers. RADIANT addresses this problem by directly minimizing the distributional shift of activations compared to its original state, which is clearly observed in Problem (3).
> > > >
> > > > Second, they are not concerned about whether the post-intervention activations lie in desirable areas. We solve this issue by establishing a probabilistic guarantee for the extent of post-intervention activations in the desirable areas. This is incorporated in the constraint of Problem (3).
> > > >
> > > > With such improvements from both practical and theoretical aspects, our proposed methods outperform other baselines, which is apparent from empirical experiments.
> > > >
> > > > **Counting all significant contributions listed above, we respectfully disagree with your opinion that our paper just follows the ITI paper. Our paper not only indicates serious problems in existing intervention works but also proposes effective ways to accommodate them.**

---

> ### Author Response · Authors · 2024-12-01
> **Response to Reviewer KLQW (4/n)**
>
> > Moreover, there can be more comparisons between experiments on different LLMs. For example, how does the performance of RADIANT change when the model size increases, and how do other baselines behave?
>
> We appreciate your suggestion on improving the depth of our paper. To accommodate your concern, we add another analysis on how RADIANT and other baselines' performance changes when gradually increasing the model size. Particularly, the table shows the baselines' Truth(%) when increasing the model size.
>
>
> | Model               | Unintervened | ITI | FSP | RADIANT | FSP + ITI | FSP + RADIANT |
> |--------------------------|------------------|---------|---------|-------------|---------------|--------------------|
> | GPT-2 Large (700M)  | 21.91            | 31.09   | 27.30   | 38.73       | 30.45         | **40.41**         |
> | Gemma-2B (2B)       | 51.23            | 54.74   | 42.23   | 59.62       | 61.57         | **64.71**         |
> | Llama-7B (7B)       | 22.16            | 28.03   | 39.78   | 44.48       | 45.16         | **57.43**         |
> | Llama3-8B (8B)      | 44.18            | 46.88   | 39.78   | 50.82       | 45.16         | **52.02**         |
> | Llama2-chat-13B (13B)| 59.86            | 63.04   | 58.63   | **74.20**   | 59.24         | 67.75         |
>
> **The table shows that RADIANT’s performance consistently improves as model size increases, demonstrating its scalability and ability to make the most of larger models.** For example, its Truth score jumps from 38.73% with GPT-2 Large (700M) to an impressive 74.20% with Llama2-chat-13B (13B). This upward trend highlights how well RADIANT adapts to the growing capacity of bigger models. In comparison, other baselines show mixed results. ITI improves slightly with larger models but remains behind RADIANT, while FSP performs decently on smaller models but doesn’t scale as effectively, sometimes even plateauing. Combining methods like FSP + ITI brings some improvement, but it still doesn’t match RADIANT’s performance. Interestingly, combining FSP with RADIANT consistently boosts results. For instance, with Llama2-chat-13B, FSP + RADIANT achieves 67.75%.. In summary, RADIANT stands out by scaling effectively with model size and outperforming other baselines across the board, especially as models grow larger.

---

> ### Author Response · Authors · 2024-12-01
> **Response to Reviewer KLQW (5/n)**
>
> > However, I believe that RADIANT is not targeted at a specific model, so it should be valid for at least 2-3 different LLMs to verify that this method works.
>
> Thank you for your constructive comment. Following your suggestion, we study the performance of finetuning-free techniques, including ITI, RADIANT, and FSP, on Gemma-2B and GPT-2 Large, which serve as alternative base models to the Llama model family. Table 3 below shows that RADIANT using few-shot prompting outperforms other methods by a large gap. Notably, FSP + RADIANT is superior to FSP + ITI in terms of both True * Info and True and MC1 scores. Concurrently, RADIANT, implemented separately, outperforms ITI and FSP in terms of True * Info and True scores while only slightly behind in MC1 and MC2.
>
> *Table 3: Quantitative results of different intervention methods on TruthfulQA dataset, across different Language Models Parameters of RADIANT: \( \alpha = 2.5, \Gamma = 15 \).*
>
> *a. Gemma-2B*
>
> | Methods | True * Info (\%) ↑ | True (%) ↑ | MC1 ↑ | MC2 ↑ | CE ↓ | KL ↓ |
> | --- | --- | --- | --- | --- | --- | --- |
> | Unintervened | 31.00 | 51.23 | 27.12 | 43.62 | 2.55 | 0.00 |
> | ITI | 33.42 | 54.74 | 29.14 | 46.01 | 2.64 | 0.17 |
> | FSP | 34.92 | 42.23 | **35.10** | **49.24** | 2.55 | 0.0 |
> | RADIANT (ours) | **35.62** | **59.62** | 30.34 | 48.06 | 2.62 | 0.15 |
> | FSP + ITI | 48.83 | 61.57 | 38.27 | 54.73 | 2.69 | 0.16 |
> | FSP + RADIANT (ours) | **56.14** | **64.71** | **39.54** | **56.98** | 2.65 | 0.09 |
>
> *b. GPT-2 Large*
>
> | Methods | True * Info (\%) ↑ | True (%) ↑ | MC1 ↑ | MC2 ↑ | CE ↓ | KL ↓ |
> | --- | --- | --- | --- | --- | --- | --- |
> | Unintervened | 19.2 | 21.91 | 23.57 | 40.75 | 2.8 | 0.0 |
> | ITI | 26.94 | 31.09 | 24.68 | **42.31** | 2.94 | 0.13 |
> | FSP | 21.82 | 27.30 | **25.34** | 42.07 | 2.8 | 0.0 |
> | RADIANT (ours) | **30.18** | **38.73** | 25.14 | 42.14 | 2.92 | 0.12 |
> | FSP + ITI | 29.53 | 30.45 | 25.12 | **44.79** | 2.98 | 0.18 |
> | FSP + RADIANT (ours) | **35.36** | **40.41** | **26.18** | 44.29 | 2.94 | 0.16 |
>
> > I think experiments provided in this paper lacks depth. I really appreciate the substantial experimental results provided by authors. However, there lacks further analysis and discussions.
>
> Thanks for your comment. We are sorry you are dissatisfied with our discussions and analysis. **However, could you kindly be more specific on which further analysis and discussions we need to provide?** The discussion period will last for another three days, and we are confident we can meet your further requests.
>
> We note that, regarding your two previous concerns, we have already added them to our revised PDF version to deepen the discussion and analysis:
>
> - Particularly, we incorporate our answers to your concern about our motivation, intuition, and what makes our framework stand out from other baselines in Section 3.
> - Additional discussions on empirical results are included in the Appendix and Section 4. With the current version, we believe that our experimental analysis and discussions are extensive and cover almost all aspects. Our proposal performs better than various state-of-the-art baselines in the QA tasks across all model sizes that our limited resources allow us to test.  Even in the toxicity mitigation task, our method is also promising compared to state-of-the-art methods like DExperts and MiMiC.
> - We also show that our method not only works for the Llma model family but also be effective for other base models like GPT and Gemma. In the scenarios, base models that have been finetuned before like Vicuna, Alpaca, and Lofit ; our proposal also performs well, showing its broad applicability. In terms of a thorough analysis of the behavior of RADIANT, we write a special part in Appendix A to delve deeper into it.
> - For each part, we all commented and analyzed the trend carefully.
>
> With all the arguments, we kindly ask the reviewer to reconsider their scores.

---

> > ### Author Response · Authors · 2024-12-02
> >
> > Dear Reviewer KLQW,
> >
> > Once again, we thank you for your helpful and constructive feedback, which definitely helps improve the quality of our paper. For your latest concerns, we have uploaded arguments and answers to address them. We look forward to knowing whether you are satisfied with them and whether they are persuasive enough to change your opinion about our paper, at least to avoid rejection evaluation. We would genuinely appreciate it if you have any further feedback on our paper. We will do our best to address them within the remaining short period of the rebuttal phase.
> >
> > Best regards, The authors.

---

### Official Review · Reviewer_P1YY · 2024-11-04

**Soundness:** 3
**Presentation:** 3
**Contribution:** 3
**Rating:** 8
**Confidence:** 4

**Summary:**

The paper addresses challenge of guiding language generation away from undesirable outputs (e.g., non-truthful responses) towards more desirable ones (e.g., truthful responses). The proposed method tackles this by intervening at the head level within a transformer model. This involves: (1) training a linear classifier to detect when an undesirable output is being generated for each head, and (2) applying an affine transformation to shift these detected head's outputs towards desirable ones. The classifier is trained using supervised learning, with a hyperparameter to balance false-positive and false-negative. Then, the method employs a distribution-matching loss to ensure that the transformation minimally diverges from the original activation while ensuring that the classifier (trained in (1)) confidently identifies activation vectors as 'desirable'. This approach tradeoffs the degree of the shift introduced and the confidence of the classifier.

**Strengths:**

(1) Interesting and significant application.
(2) Introduce a trade-off between preserving original information and matching the 'desirable' distribution makes sense to me.
(3) Strong results on QA

**Weaknesses:**

I mix here aspects which I perceive as weakness is some aspects which are not entirely clearly to me and which I would like authors to clarify

1) The proposed RADIANT approach builds on ITI by Li et al. (2023), and the differences are the 'risk-aware' optimization for learning probes, the constraint that affected heads are in the same layer, and the specific optimization objective. The first two differences are clear, and the ablation studies suggest they have tangible benefits. However, I’m less certain about the intuition behind the optimization objective itself. For instance,  I understand that the step size in the 'truthfulness direction' as defined in ITI might not always be appropriate (e.g., it’s unclear why that specific size is optimal or whether a good hyperparameter alpha can be consistently set across all heads) and the idea of taking the smallest step to meet the probabilistic constraint (as in RADIANT) aligns with my intuition; I find it compelling.  But the part I find puzzling is the requirement to match as closely the covariance of the original 'undesirable' (non-intervened) activations with the 'corrected'  activations. This objective seems to differ fundamentally from a similar regularization term in MiMiC by Singh et al. (2024), which aims to match intervened data with 'desirable' data. There, the goal is clear: it ensures that the data distributions are indistinguishable by any classifier; this makes sense to have this constraint on the distribution level. Here, though, I don’t quite understand why enforcing a distributional similarity between the undesirable activations and intervened steps is better than simply making the smallest step necessary for every activation (with the step size define according to the probabilistic constraint). Could the authors clarify the rationale behind this assumption?

(2) The paper includes comparisons with ITI but lacks an explicit comparison with MiMiC. Given that MiMiC has been available on arXiv for 7 months before the ICLR 2025 deadline, such analysis may be expected. The authors claim that MiMiC does not preserve semantics (and I tend to believe i), but I would like to see empirical confirmation of this in the paper.

(3)  The evaluation primarily focuses on QA tasks (TrustfulQA in the main paper + two other QA datasets in appendix).  Is not the idea more broad than that? I would have like the approach to be applied to a broader range of generation tasks, and types of interventions.

(4)  MiMiC ensures that a broad class of classifiers cannot distinguish post-intervention activations from desirable ones, implicitly implying that furrther layers cannot see the differnce. In contrast, this method optimizes protection against only specific classifier trained in the first stage. Is this a limitation? Consequently, it’s possible that a different classifier could still distinguish the activations. Could this  affect the robustness of the intervention?

(5) Why uniformly weighting divergences in mean and variance in objective (3). Why this specific way of measuring the divergence between distributions?  (Though I have a more general question, why define this objective in terms of proabilities)

(6) Section 3 is a bit dense, and giving some intuitions up-front would make it a more enjoyable read.

**Questions:**

See above, all my comments are at the same time questions.

---

> ### Author Response · Authors · 2024-11-22
> **Thank you for your valuable feedback and appreciation of our work**
>
> We thank the reviewer for the valuable feedback and appreciation of our work. Here are our replies to your questions.
>
> >  I’m less certain about the intuition behind the optimization objective itself. The part I find puzzling is the requirement to match as closely the covariance of the original 'undesirable' (non-intervened) activations with the 'corrected' activations. This objective seems to differ fundamentally from a similar regularization term in MiMiC by Singh et al. (2024), which aims to match intervened data with 'desirable' data.
>
> Thank you for your thoughtful question. We think that there is likely a misunderstanding about this aspect of the paper. Indeed, we want to closely match the covariance of the post-intervention activations, not the original 'undesirable' ones, with the 'corrected' activations. We note that despite the two seemingly different learning objectives for the intervention, the high-level ideas of both MIMIC and our paper are the same.
>
> In particular, our second criterion, in which we want the intervention to be effective in converting the undesirable activations to the desirable regions, is the same as the objective of MiMiC (aiming to match intervened data with empirically observed ‘desirable’ data distribution). This is reflected in the constraint of the optimization problem (3).
>
> However, different from MiMiC, we argue that the intervention needs to match criterion (iii): minimize the magnitude of the intervention to sustain the context of the input – so that the output remains semantically correct.
>
> >  I don’t quite understand why enforcing a distributional similarity between the undesirable activations and intervened steps is better than simply making the smallest step necessary for every activation (with the step size defined according to the probabilistic constraint). Could the authors clarify the rationale behind this assumption?
>
> We argue that solving the optimization problem in Eq. (3) achieves exactly what the reviewer suggested. The goal is to identify an intervention that pushforwards the distribution of undesirable activations into the desirable region. However, this intervention must be small enough to avoid moving the distribution too far from its original state.
>
> > The evaluation primarily focuses on QA tasks (TrustfulQA in the main paper + two other QA datasets in appendix). Isn't the idea more broad than that?
>
> Thank you for your thorough and constructive suggestion. We have conducted an additional experiment to verify the effectiveness of RADIANT on the toxicity mitigation task. The details of this experiment are available in Appendix 4.1, which is newly added in our revised PDF version. Here, we briefly present the experimental setting for the experiment and our results.
>
> We use the Toxic Comments Classification Challenge dataset [1] with preprocessing from the MiMiC paper [2].
> GPT-2 Large serves as the base model for our experiments in this task to have a fair comparison with previous works.
>
> We report two metrics to measure the toxicity scores: Expected Maximum Toxicity, denoted as Exp. Max. Tox., and Toxic Completion Proportion abbreviated as Tox. Prob. We also include Fluency, which is evaluated by calculating the perplexity of the generated outputs using GPT-2 (XL) as a reference model. Moreover, we compute the diversity score assessed by examining the ratio of unique n-grams (1-gram, 2-gram, and 3-gram) to the total number of tokens in the generated text. It is worth noting that these metrics are main and common in previous papers.
>
> We include several baselines that have the same goal of reducing the toxicity of LLMs, including MiMiC [2], DEXPERTS [3], DAPT [4], UDDIA [5], PPLM [6], GOODTRIEVER [7]. As for MIMIC, we consider two versions: Mean Matching (MM) and Mean+Covariance Matching (MCM). Both these versions are introduced in their original paper.
> The experimental result is presented in the below table.

---

> ### Author Response · Authors · 2024-11-22
> **Reponse to Reviewer P1YY (2/3)**
>
> | Model            | Exp. Max. Tox. ↓ | Tox. Prob. ↓ | Fluency ↓ | 1-gram ↑ | 2-gram ↑ | 3-gram ↑ |
> |-------------------|------------------|--------------|-----------|----------|----------|----------|
> | GPT-2 (large)     | 0.39            | 0.25         | 24.66     | 0.58     | 0.85     | 0.85     |
> | **DAPT**          | 0.27            | 0.09         | 30.27     | 0.57     | 0.84     | 0.84     |
> | **GeDI**          | 0.24            | 0.06         | 48.12     | 0.62     | 0.84     | 0.83     |
> | **PPLM (10%)**    | 0.38            | 0.24         | 32.58     | 0.58     | **0.86** | **0.86** |
> | **UDDIA**         | 0.24            | 0.04         | **26.83** | 0.51     | 0.80     | 0.83     |
> | **DExperts**      | **0.21**        | **0.02**     | 27.15     | 0.56     | 0.84     | 0.84     |
> | **GOODTRIEVER**   | 0.22            | 0.04         | 27.11     | 0.58     | 0.82     | 0.83     |
> | **MM (MiMiC)**    | 0.33            | 0.16         | 28.00     | **0.58** | **0.85** | **0.85** |
> | **MCM (MiMiC)**   | 0.29            | **0.09**     | 30.70     | 0.54     | 0.84     | 0.84     |
> | **ITI**           | 0.31            | 0.12         | 33.12     | 0.57     | **0.85** | **0.85** |
> | **RADIANT**       | **0.27**        | **0.09**     | **27.10** | **0.58** | **0.85** | **0.85** |
>
> Baselines are divided into two groups: those requiring extensive fine-tuning or gradient computations during inference(e.g., DAPT, GeDI, PPLM, UDDIA, DExperts, GOODTRIEVER) and light-weight inference-time methods (e.g., MIMIC, ITI, RADIANT). While the fine-tuning group generally performs better on toxicity metrics, these methods are computationally expensive due to the necessity of fine-tuning or gradient computations during inference.
>
> Inference-time methods like MIMIC, ITI, and RADIANT achieve comparable toxicity reduction to fine-tuning methods but are significantly more resource-efficient. Among these, RADIANT stands out by delivering the best toxicity reduction while also maintaining strong fluency and diversity. In fact, RADIANT’s fluency surpasses nearly all methods in the fine-tuning group except UDDIA, and its diversity is on par with the best (PPLM). This highlights RADIANT’s balance of efficiency and performance.
>
> > The paper includes comparisons with ITI but lacks an explicit comparison with MiMiC.
> On the new Toxic Comments Classification Challenge benchmark, we compared RADIANT with MiMiC. While both achieve similar performance in Tox. Prob. and diversity metrics, RADIANT significantly outperforms MiMiC in Fluency and Exp. Max. Tox. This highlights RADIANT's effectiveness and superiority in the task.
>
> >  MiMiC ensures that a broad class of classifiers cannot distinguish post-intervention activations from desirable ones, implicitly implying that further layers cannot see the difference. In contrast, this method optimizes protection against only specific classifiers trained in the first stage. Is this a limitation? It’s possible that a different classifier could still distinguish the activations. Could this affect the robustness of the intervention?
>
> It is possible that a different classifier could still distinguish the activations. However, different from MiMiC, it ensures that a broad class of classifiers cannot distinguish post-intervention activations from desirable ones; in our RADIANT framework, we do not have any similar constraint or objective terms to ensure that no classifier could still distinguish the post-intervention activations and the human-labeled desired ones. Instead of that, we argue that this condition is not necessary to ensure truthfulness. This is because the empirical distributions of desired samples are approximated using a limited number of samples, while the actual distributions span a much broader space.
>
> An intuitive example is as follows. We denoted A as our desired activations and B as activations of their paraphrase to passive expression. By common sense, we can use the classifier that distinguished activate and passive tone as a good candidate classifier for this case. But A and B still offer truthful answers.
>
> We conclude by arguing that this is not our limitation. This is the difference between our approach to the problem and the MiMiC paper. In practice, although RADIANT does not consider this condition, the extra experimental benchmarks show that our framework still performs better than MiMiC, which ensures this condition. Our response to your questions above discusses the comparison between RADIANT and MiMiC.

---

> ### Author Response · Authors · 2024-11-22
> **Reponse to Reviewer P1YY (3/3)**
>
> > Why uniformly weighting divergences in mean and variance in objective (3). Why this specific way of measuring the divergence between distributions? (Though I have a more general question: why define this objective in terms of probabilities)
>
> We define this objective in terms of probabilities as an inspiration from the Bures-Wasserstein distance, an optimal transport problem that finds an optimal matching between two Gaussian distributions. Indeed, we re-emphasize that from the optimization objective (3), one can have the freedom to choose different types of divergence, different from the one we proposed in Theorem 1. This showcases the flexibility of our framework in solving the intervention steps.
>
> >Section 3 is a bit dense, and giving some intuitions up-front would make it a more enjoyable read.
>
> Thank you for this constructive comment on our presentation. In our revised version, we added a paragraph outlining our intuition at the beginning of Section 3. We also revised the three beginning paragraphs of Section 3 to make it less notation-dense.
>
> **Reference**
>
> [1] https://huggingface.co/openai-community/gpt2-large
>
> [2] Singh, Shashwat, et al. "Representation Surgery: Theory and Practice of Affine Steering." Forty-first International Conference on Machine Learning.
>
> [3] Liu, Alisa, et al. "DExperts: Decoding-time controlled text generation with experts and anti-experts." arXiv preprint arXiv:2105.03023 (2021).
>
> [4] Gururangan, Suchin, et al. "Don't stop pretraining: Adapt language models to domains and tasks." arXiv preprint arXiv:2004.10964 (2020).
>
> [5] Yang, Zonghan, et al. "Unified detoxifying and debiasing in language generation via inference-time adaptive optimization." arXiv preprint arXiv:2210.04492 (2022).
>
> [6] Dathathri, Sumanth, et al. "Plug and play language models: A simple approach to controlled text generation." arXiv preprint arXiv:1912.02164 (2019).
>
> [7] Pozzobon, Luiza, et al. "Goodtriever: Adaptive toxicity mitigation with retrieval-augmented models." arXiv preprint arXiv:2310.07589 (2023).

---

> > ### Comment · Reviewer_P1YY · 2024-11-29
> > **thank you for the changes and clarifications**
> >
> > Thanks for the clarification (including clearing up my confusion about the distribution matching constraint), the revisoin of Section 3, and the extra experiments on the Toxic Comments Challenge. Apologies for taking so long to re-read the paper -- I’m raising my score!

---

> > > ### Author Response · Authors · 2024-11-29
> > >
> > > Thank you for your appreciation of our work. We think that the feedback from yours and other reviewers helps improve our submission a lot.

---

### Author Response · Authors · 2024-11-22
**[Common Response] Thank you for your thoughtful reviews**

We thank the reviewers for their valuable feedback and comments. We have incorporated these feedbacks into the revised version of our work: in the updated PDF, we have highlighted in red all the modifications that we have made to our submission.

We are pleased to note that there is a consensus among reviewers regarding our paper's contribution: it presents **a novel intervention strategy to mitigate undesirable content generation, which is a significant research problem**, along with **interesting and significant applications,** and demonstrates **strong results on the QA dataset**.

Besides that, a common request among the reviewers is extra empirical benchmarks with different tasks and families of models. Hence, we have performed the required experiments, and following this thread are the results that we have done. (We will also make separate replies that address the concerns of each reviewer in their review section.)

---

> ### Author Response · Authors · 2024-11-22
> **Toxicity Mitigation Task**
>
> We benchmark the performance of RADIANT in mitigating toxicity in long-form text generation. In this task, large language models (LLMs) are required to complete an incomplete prefix of text. Typically, the prefix prompt is designed to elicit toxic content from LLMs. To ensure a fair comparison with previous works, we set up experiments following Singh et al (2024). and Pozzobon et al. (2023), as detailed below.
> - **Training Dataset** We use the Toxic Comments Classification Challenge data, which comprises sentences and their human toxicity labels. The data preprocessing follows Singh et al., while the gathering of activations is identical to the procedure used in the QA task.
>
> - **Models** Following existing works in the field, we adopt GPT-2 Large as the base model across all experiments in the toxicity mitigation task.
>
> - **Hyperparameters** As mentioned in the QA task section, there are two pivotal hyperparameters in the RADIANT framework: $ \alpha $ and $ \Gamma = \Phi^{-1}(1-\gamma) $. These will be selected through a grid search procedure detailed in Appendix A.
>
> - **Baselines** We include several baselines that aim to reduce the toxicity of LLMs, including  MIMIC (Singh et al., 2024), DEXPERTS (Liu et al., 2021), DAPT (Gururangan et al., 2020), UDDIA (Yang et al., 2022), PPLM (Dathathri et al., 2019), GOODTRIEVER (Pozzobon et al., 2023). For MIMIC, we consider two versions: Mean Matching (MM) and Mean+Covariance Matching (MCM), as introduced in their original paper.
>
> - **Metrics** We assess the performance of the models using three key metrics: toxicity, fluency, and diversity, outlined as follows:
>
>   - **Toxicity**: We use the non-toxic split of RealToxicityPrompts and utilize the evaluation framework from Liu et al. and Singh et al. For each prompt in the dataset, the models generate 25 outputs, each capped at 20 tokens in length. The parameters of the shared decoding mechanism for all algorithms are presented in Table 1. These outputs are analyzed using the Perspective API, which estimates the likelihood that a human would perceive the text as toxic. Two metrics are derived:
> 	- **Expected Maximum Toxicity** (denoted as Exp. Max. Tox.): For every prompt, we identify the output with the highest toxicity score and compute the average of these maximum scores across all prompts.
> 	- **Toxic Completion Proportion** (abbreviated as Tox. Prob.): This metric tracks the fraction of outputs considered toxic, defined as having a score above 0.5 based on the Perspective API's threshold.
>    - **Fluency**: We evaluate fluency by calculating the perplexity of the generated outputs using GPT-2 (XL) as a reference model. Lower perplexity values indicate more coherent and grammatically fluent text.
>   - **Diversity**: We assess diversity by examining the ratio of unique n-grams (1-gram, 2-gram, and 3-gram) to the total number of tokens in the generated text. This metric captures variation in outputs, with higher values indicating more diverse language use.
>
> References:
>
> - Singh, Shashwat, et al. (2024) "Representation Surgery: Theory and Practice of Affine Steering." Forty-first International Conference on Machine Learning.
> - Luiza Pozzobon, Beyza Ermis, Patrick Lewis, and Sara Hooker. Goodtriever: Adaptive toxicity mitigation with retrieval-augmented models. arXiv preprint arXiv:2310.07589, 2023.
> - Alisa Liu, Maarten Sap, Ximing Lu, Swabha Swayamdipta, Chandra Bhagavatula, Noah A Smith, and Yejin Choi. Dexperts: Decoding-time controlled text generation with experts and anti-experts. arXiv preprint arXiv:2105.03023, 2021.
> - Suchin Gururangan, Ana Marasovi´c, Swabha Swayamdipta, Kyle Lo, Iz Beltagy, Doug Downey, and Noah A Smith. Don’t stop pretraining: Adapt language models to domains and tasks. arXiv preprint arXiv:2004.10964, 2020
> - Zonghan Yang, Xiaoyuan Yi, Peng Li, Yang Liu, and Xing Xie. Unified detoxifying and debiasing in language generation via inference-time adaptive optimization. arXiv preprint arXiv:2210.04492, 2022
> - Sumanth Dathathri, Andrea Madotto, Janice Lan, Jane Hung, Eric Frank, Piero Molino, Jason Yosinski, and Rosanne Liu. Plug and play language models: A simple approach to controlled text generation. arXiv preprint arXiv:1912.02164, 2019.

---

> ### Author Response · Authors · 2024-11-22
> **Toxicity Mitigation Task (continued)**
>
> - **Results** The experimental results for baselines are shown in Table 1, where all methods utilize GPT-2 Large as their base model. The result for the original model is described in the first row. We categorize baselines into two groups:
>
> 1. The first group uses an extensive fine-tuning procedure and includes DAPT, GeDI, PPLM, UDDIA, DExperts, and GOODTRIEVER.
> 2. The second group contains inference-time fine-tuning-free methods like MIMIC, ITI, and RADIANT.
>
> Baselines in the first group outperform those in the second group regarding toxicity metrics; however, they require either fine-tuning or computing gradients during inference time, which can be computationally intensive. MIMIC, ITI, and RADIANT achieve comparable toxicity reduction to many algorithms in the first group while consuming significantly fewer resources. **Specifically, RADIANT outperforms PPLM and is equally competitive with DAPT. Notably, within the second group, RADIANT provides superior toxicity reduction compared to ITI and MIMIC while maintaining better fluency and diversity in generated sentences. The fluency of RADIANT is favored over nearly all algorithms in the first group except for UDDIA. Additionally, its diversity metric surpasses that of other baselines apart from PPLM.**
>
> *Table 1: Quantitative results of different intervention methods on RealToxicityPrompts dataset.* Parameters of RADIANT: $ \alpha = 2 .5 , \Gamma =15 $.
>
> | Model | Exp. Max. Tox. ↓ | Tox. Prob. ↓ | Fluency ↓ | 1-gram ↑ | 2-gram ↑ | 3-gram ↑ |
> | --- | --- | --- | --- | --- | --- | --- |
> | GPT-2 (large) | 0.39 | 0.25 | 24.66 | 0.58 | 0.85 | 0.85 |
> | DAPT | 0.27 | 0.09 | 30.27 | 0.57 | 0.84 | 0.84 |
> | GeDI | 0.24 | 0.06 | 48.12 | 0.62 | 0.84 | 0.83 |
> | PPLM (10%) | 0.38 | 0.24 | 32.58 | 0.58 | **0.86** | **0.86** |
> | UDDIA | 0.24 | 0.04 | **26.83** | 0.51 | 0.80 | 0.83 |
> | DExperts | **0.21** | **0.02** | 27.15 | 0.56 | 0.84 | 0.84 |
> | GOODTRIEVER | 0.22 | 0.04 | 27.11 | 0.58 | 0.82 | 0.83 |
> | MM (MimiC) | 0.33 | 0.16 | 28.00 | **0.58** | **0.85** | **0.85** |
> | MCM (MimiC) | 0.29 | **0.09** | 30.00 | 0..54 | 0.84 | 0..84 |
> | ITI | 0.31 | 0.12 | 33.12 | 0.57 | **0.85** | **0.85** |
> | RADIANT | **0.27** | **0.09** | **27.10** | **0.58** | **0.85** | **0.85** |

---

> ### Author Response · Authors · 2024-11-22
> **New baseline added to benchmark of finetuning-free methods (Section 4.2.1)**
>
> We have added two new baselines into Section 4.2.1, which are NL-ITI (Hoscilowicz et al., 2024) and LITO (Bayat et al., 2024). We notice that our framework RADIANT still consistently returned the highest True*Info and True metric, both when running alone and combining with FSP.
>
> Table 2. Quantitative results of different intervention methods on TruthfulQA dataset, across different Language Models. Parameters of RADIANT: $\alpha = 2.5, \Gamma = 15$.
>
> 1. Llama-7B
>
> | Methods | True * Info (\%) ↑ | True (%) ↑ | MC1 ↑ | MC2 ↑ | CE ↓ | KL ↓ |
> | --- | --- | --- | --- | --- | --- | --- |
> | Unintervened         |            21.15            |         22.16        |      25.58     |      40.54     |       2.13      |       0.00      |
> | ITI                  |            26.52            |         28.03        |      27.78     |      43.59     |       2.20      |       0.07      |
> | FSP                  |            36.13            |         39.78        |    **34.03**   |    **50.34**   |       2.13      |       0.00      |
> | NL-ITI               |            29.06            |         38.04        |      32.97     |      45.69     |       2.19      |       0.07      |
> | LITO                 |            39.08            |         41.22        |      29.22     |      47.64     |       2.19      |       0.07      |
> | RADIANT (ours)       |          **40.36**          |       **44.48**      |      30.91     |      46.13     |       2.19      |       0.07      |
> | FSP + ITI            |            40.63            |         45.16        |      35.50     |      52.48     |       2.20      |       0.07      |
> | FSP + NL-ITI         |            45.97            |         47.31        |    **38.37**   |      53.61     |       2.20      |       0.07      |
> | FSP + LITO           |            49.05            |         55.68        |      36.23     |      54.92     |       2.20      |       0.07      |
> | FSP + RADIANT (ours) |          **49.31**          |       **57.43**      |      37.97     |    **55.31**   |       2.20      |       0.08      |
>
> b) Llama3-8B
>
> | Methods | True * Info (\%) ↑ | True (%) ↑ | MC1 ↑ | MC2 ↑ | CE ↓ | KL ↓ |
> | --- | --- | --- | --- | --- | --- | --- |
> | Unintervened         |            32.88            |         44.18        |      30.36     |      48.98     |       2.38      |       0.00      |
> | ITI                  |            35.92            |         46.88        |      32.07     |      49.84     |       2.50      |       0.13      |
> | FSP                  |            36.32            |         39.78        |    **35.74**   |      52.93     |       2.38      |       0.00      |
> | NL-ITI               |            35.98            |         45.72        |      33.02     |      51.37     |       2.50      |       0.13      |
> | LITO                 |            37.53            |         48.20        |      34.96     |      52.54     |       2.48      |       0.11      |
> | RADIANT (ours)       |          **37.78**          |       **50.82**      |      33.82     |    **52.98**   |       2.48      |       0.08      |
> | FSP + ITI            |            40.63            |         45.16        |      35.50     |      52.98     |       2.48      |       0.14      |
> | FSP + NL-ITI         |            40.70            |         46.03        |      34.15     |      53.35     |       2.49      |       0.14      |
> | FSP + LITO           |            43.95            |         49.82        |    **38.41**   |    **55.31**   |       2.54      |       0.17      |
> | FSP + RADIANT (ours) |          **44.09**          |       **52.02**      |      37.98     |      54.61     |       2.52      |       0.15      |
>
> c) Llama2-chat-13B
>
> | Methods | True * Info (\%) ↑ | True (%) ↑ | MC1 ↑ | MC2 ↑ | CE ↓ | KL ↓ |
> | --- | --- | --- | --- | --- | --- | --- |
> | Unintervened | 51.87 | 59.86 | 35.38 | 53.32 | 2.31 | 0.00 |
> | ITI | 57.02 | 63.04 | 37.46 | 55.59 | 2.32 | 0.17 |
> | FSP | 55.97 | 58.63 | **40.76** | 57.84 | 2.31 | 0.00 |
> | NL-ITI | 57.13 | 60.82 | 39.01 | 57.24 | 2.33 | 0.17 |
> | LITO | 58.12 | 61.36 | 38.25 | 57.21 | 2.34 | 0.18 |
> | RADIANT (ours) | **63.68** | **74.20** | 39.95 | **58.18** | 2.35 | 0.18 |
> | FSP + ITI | 56.78 | 59.24 | 41.50 | 59.01 | 2.33 | 0.13 |
> | FSP + NL-ITI | 59.62 | 61.77 | 42.15 | 57.87 | 2.34 | 0.15 |
> | FSP + LITO | 60.74 | 63.21 | 41.28 | 58.46 | 2.36 | 0.17 |
> | FSP + RADIANT (ours) | **64.68** | **67.75** | **42.52** | **59.99** | 2.38 | 0.18 |

---

> ### Author Response · Authors · 2024-11-22
> **The effectiveness of RADIANT beyond the LLAMA base models**
>
> We study the performance of finetuning-free techniques, including ITI, RADIANT, and FSP, on Gemma-2B and GPT-2 Large, which serve as alternative base models to the Llama model family. Table 3 below shows that RADIANT using few-shot prompting outperforms other methods by a large gap. Particularly, FSP + RADIANT enhances the True * Info score of Gemma-2B and GPT-2 Large by 25.14\% and 16.16\%, respectively. Notably, FSP + RADIANT is superior to FSP + ITI in terms of both True * Info and True and MC1 scores. Concurrently, RADIANT, implemented separately, outperforms ITI and FSP in terms of True * Info and True scores while only slightly behind in MC1 and MC2.
>
> *Table 3: Quantitative results of different intervention methods on TruthfulQA dataset, across different Language Models Parameters of RADIANT: \( \alpha = 2.5, \Gamma = 15 \).*
>
> 1. *Gemma-2B*
>
> | Methods | True * Info (\%) ↑ | True (%) ↑ | MC1 ↑ | MC2 ↑ | CE ↓ | KL ↓ |
> | --- | --- | --- | --- | --- | --- | --- |
> | Unintervened | 31.00 | 51.23 | 27.12 | 43.62 | 2.55 | 0.00 |
> | ITI | 33.42 | 54.74 | 29.14 | 46.01 | 2.64 | 0.17 |
> | FSP | 34.92 | 42.23 | **35.10** | **49.24** | 2.55 | 0.0 |
> | RADIANT (ours) | **35.62** | **59.62** | 30.34 | 48.06 | 2.62 | 0.15 |
> | FSP + ITI | 48.83 | 61.57 | 38.27 | 54.73 | 2.69 | 0.16 |
> | FSP + RADIANT (ours) | **56.14** | **64.71** | **39.54** | **56.98** | 2.65 | 0.09 |
>
> *b. GPT-2 Large*
>
> | Methods | True * Info (\%) ↑ | True (%) ↑ | MC1 ↑ | MC2 ↑ | CE ↓ | KL ↓ |
> | --- | --- | --- | --- | --- | --- | --- |
> | Unintervened | 19.2 | 21.91 | 23.57 | 40.75 | 2.8 | 0.0 |
> | ITI | 26.94 | 31.09 | 24.68 | **42.31** | 2.94 | 0.13 |
> | FSP | 21.82 | 27.30 | **25.34** | 42.07 | 2.8 | 0.0 |
> | RADIANT (ours) | **30.18** | **38.73** | 25.14 | 42.14 | 2.92 | 0.12 |
> | FSP + ITI | 29.53 | 30.45 | 25.12 | **44.79** | 2.98 | 0.18 |
> | FSP + RADIANT (ours) | **35.36** | **40.41** | **26.18** | 44.29 | 2.94 | 0.16 |

---

> > ### Author Response · Authors · 2024-11-29
> > **Comparison of RADIANT vs. supervised fine-tuning (SFT)**
> >
> > Following Reviewer CbQ1's suggestion, we provide below the results for supervised fine-tuning comparison benchmark
> >
> > **SFT on GPT2-Large**
> >
> > Due to computational constraints and SFT's requirement to finetune all LLM parameters that demands substantial GPU resources for backpropagation operation, we can only perform SFT on the GPT2-large, the smallest model in our experiments. The results are available in Table 1 below. Once again, we want to highlight the advantages of inference time methods like ours: by avoiding gradient computation or backpropagation, they offer a lightweight, fast, versatile, and economical way to improve the performance of LLMs. This is especially useful in low-resource scenarios.
> >
> > *Table 1: Quantitative results of different intervention methods on TruthfulQA dataset on GPT2-Large. Parameters of RADIANT: $\alpha = 2.5, \Gamma = 15$.*
> > | Methods | True * Info (%) ↑ | True (%) ↑ | MC1 ↑ | MC2 ↑ | CE ↓ | KL ↓ |
> > |---------|------------------|-------------|--------|--------|-------|-------|
> > | Unintervened | 19.20 | 21.91 | 23.57 | 40.75 | 2.8 | 0.0 |
> > | SFT | **35.16** | 38.28 | **35.70** | **53.57** | 3.27 | 0.46 |
> > | ITI | 26.94 | 31.09 | 24.68 | 42.31 | 2.94 | 0.13 |
> > | FSP | 21.82 | 27.30 | 25.34 | 42.07 | 2.8 | 0.0 |
> > | RADIANT (ours) | 30.18 | **38.73** | 25.14 | 42.14 | 2.92 | 0.12 |
> > | FSP + ITI | 29.53 | 30.45 | 25.12 | **44.79** | 2.98 | 0.18 |
> > | FSP + RADIANT (ours) | **35.36** | **40.41** | **26.18** | 44.29 | 2.94 | 0.16 |
> >
> > **SFT on Llama-7B**
> >
> > Because Llama-7B is used as a base model for many of our experiments, we also include the results SFT on Llama-7B for comparison, but it is worth noting that this result is referred from the ITI paper (Table 1, Li et al. 2024). Since our evaluation framework differs from ITI in terms of the GPT-judge and GPT-info models (which is attributed to the fact that these models in the ITI paper are no longer available in the OpenAI), the results may not be fair for comparison.
> >
> > *Table 2: Quantitative results of different intervention methods on TruthfulQA dataset on Llama-7B. Parameters of RADIANT: $\alpha = 2.5, \Gamma = 15$.*
> > | Methods | True * Info (%) ↑ | True (%) ↑ | MC1 ↑ | MC2 ↑ | CE ↓ | KL ↓ |
> > |---------|------------------|-------------|--------|--------|-------|-------|
> > | Unintervened | 21.15 | 22.16 | 25.58 | 40.54 | 2.13 | 0.00 |
> > | SFT* | 36.10 | **47.10** | 24.20 | - | 2.10 | 0.01 |
> > | ITI | 26.52 | 28.03 | 27.78 | 43.59 | 2.20 | 0.07 |
> > | FSP | 36.13 | 39.78 | **34.03** | **50.34** | 2.13 | 0.00 |
> > | RADIANT (ours) | **40.36** | **44.48** | 30.91 | 46.13 | 2.19 | 0.07 |
> > | FSP + ITI | 40.63 | 45.16 | 35.50 | 52.48 | 2.20 | 0.07 |
> > | FSP + RADIANT (ours) | **49.31** | **57.43** | 37.97 | **55.31** | 2.20 | 0.08 |
> >
> > **From Table 2, SFT achieves the best performance in terms of MC metrics and reaches a high score of True * Info and True. Regarding the True score, RADIANT still outperforms SFT in both the individual and integrating versions with FSP, offering 38.73% and 40.41% correct answers, respectively. When combined with FSP, RADIANT achieves 35.36% in True * Info score, surpassing SFT but requiring much less resources.**
> >
> > For the implementation of SFT, we use the SFTTrainer framework from Hugging Face, one of the most popular frameworks for this algorithm. While we kept the default for many parameters proposed by the library, we had to tune almost all of the important parameters like learning rate, parameters of Adam optimizer, weight decay, and so on, to get a consistent and stable fine-tuned model. Some important parameters for SFT are reported in the table below, while its best performance is represented in Table 2.
> >
> > This observation strongly supports the practicability of RADIANT, which only necessitates tuning two key hyper-parameters $\alpha$ in the probe loss, and $\Gamma = \Phi^{-1}(1-\gamma)$ in the computation of the intervention map. A thorough analysis of these parameters in an attempt to offer insights into their impact is presented in Appendix A.
> >
> > Furthermore, compared to other methods in the field, like ITI, we claim that the grid search on two hyper-parameters like ours is efficient and reasonable, so it is not harder to tune the hyper-parameters of RADIANT than other previous works.
> >
> > *Table 3: parameter for supervised fine-tuning*
> > | Parameter | Value |
> > |-----------|-------|
> > | learning_rate | 0.00002 |
> > | weight_decay | 0 |
> > | adam_beta1 | 0.8 |
> > | adam_beta2 | 0.999 |
> > | adam_epsilon | $$1 \times 10^{-8}$$ |
> > | max_grad_norm | 1 |
> > | batch_size | 32 |
> > | epochs_num | 5 |
> > | lr_scheduler_type | linear |

---

### Author Response · Authors · 2024-11-25
**Discussion deadline is coming soon**

Dear Reviewers,

The authors-reviewers discussion period is ending very soon. We would be really grateful if you could acknowledge reading the rebuttal, and are happy to answer further questions.

If you find our rebuttal answers most of your technical concerns, it will be equally great if the reviewers can consider raising their evaluations.

Best regards,
The authors.

---

### Meta-Review · Area_Chair_GcVp · 2024-12-14

**Metareview:**

This paper presents a two-stage approach to mitigating harmful content in language models. In the first stage, layer-wise classifiers identify undesirable content based on activations, while the second stage applies targeted interventions to adjust attention heads, ensuring minimal effectiveness. The proposed method outperforms baseline approaches across multiple models and datasets.

As noted by Reviewer CbQ1, this approach is incremental with limited novelty and may require extensive hyperparameter tuning.

Additionally, the experiments were limited, as Reviewers KLQW, CbQ1, and P1YY pointed out. While additional experiments were included during the rebuttal phase, they may indicate significant deviations from the original work. I recommend that the paper undergo another round of review to fully demonstrate the effectiveness of the authors' proposed method.

Therefore, I recommend rejection.

**Additional Comments On Reviewer Discussion:**

The authors have made significant efforts during the rebuttal process. However, the addition of numerous experiments suggests that the paper may require another round of review to reach its full potential.

Reviewer CbQ1 noted that the proposed method is an incremental approach with limited innovation compared to ITI and may require extensive hyperparameter tuning.

Similarly, Reviewer P1YY expressed concerns that the paper still requires further revision. The authors' responses failed to provide sufficient clarity or conviction. Specifically, the reviewer found the paper's motivation unclear, and the experiments lacked depth. While additional experiments were included, they were not accompanied by thorough analysis.

Given these concerns, I believe this work requires further revision.

---

### Decision · Program_Chairs · 2025-01-22

Reject